# Health Benefits of Plant-Based Nutrition: Focus on Beans in Cardiometabolic Diseases

**DOI:** 10.3390/nu13020519

**Published:** 2021-02-05

**Authors:** Amy P. Mullins, Bahram H. Arjmandi

**Affiliations:** 1Department of Nutrition, Food, and Exercise Sciences, Florida State University, Tallahassee, FL 32306, USA; apm2543@my.fsu.edu; 2Center for Advancing Exercise and Nutrition Research on Aging, Florida State University, Tallahassee, FL 32306, USA; 3Department of Family and Consumer Sciences--Leon County Extension Services, University of Florida Institute of Food and Agricultural Sciences, Tallahassee, FL 32301, USA

**Keywords:** CVD, type-2 diabetes, obesity, endothelial dysfunction, plant-based diet, bean, legumes, phytochemicals, SCFA

## Abstract

Cardiovascular disease (CVD) is the leading cause of death worldwide, claiming over 650,000 American lives annually. Typically not a singular disease, CVD often coexists with dyslipidemia, hypertension, type-2 diabetes (T2D), chronic system-wide inflammation, and obesity. Obesity, an independent risk factor for both CVD and T2D, further worsens the problem, with over 42% of adults and 18.5% of youth in the U.S. categorized as such. Dietary behavior is a most important modifiable risk factor for controlling the onset and progression of obesity and related disease conditions. Plant-based eating patterns that include beans and legumes support health and disease mitigation through nutritional profile and bioactive compounds including phytochemical. This review focuses on the characteristics of beans and ability to improve obesity-related diseases and associated factors including excess body weight, gut microbiome environment, and low-grade inflammation. Additionally, there are growing data that link obesity to compromised immune response and elevated risk for complications from immune-related diseases. Body weight management and nutritional status may improve immune function and possibly prevent disease severity. Inclusion of beans as part of a plant-based dietary strategy imparts cardiovascular, metabolic, and colon protective effects; improves obesity, low-grade inflammation, and may play a role in immune-related disease risk management.

## 1. Introduction

As the leading cause of morbidity and mortality, cardiovascular disease (CVD) is responsible for approximately 18 million (31%) deaths worldwide [1] and 1 of every 4 deaths annually in the United States (U.S.) [2]. Obesity is an independent risk factor for the development and progression of CVD and comorbidities including high blood pressure, dyslipidemia, and/or type-2 diabetes (T2D) [3]. Obesity continues to rise at an astonishing rate, increasing from 30% of U.S. adults to an estimated 42% over the past 20 years. Additionally, what is equally concerning is that 1 in every 5 children and adolescents are currently overweight or obese [3]. Youth that are overweight or obese have a higher likelihood of sustaining excess body weight into adulthood, with significant and premature health consequences including T2D, heart disease, cancer, and osteoarthritis [4]. Obesity is a disease associated with chronic increased cytokine production and low-grade inflammation [5,6,7]. Obese individuals have a greater risk of experiencing “cytokine storms” [5,6] and COVID-19 complications, particularly if comorbidities such as T2D or CVD are involved [8,9,10]. Although underlying reasons for the worldwide and American state of obesity are multifactorial, the link between poor diet quality, excess body weight, and disease has been clearly established.

Dietary behaviors that include more plant-derived foods and less saturated fat (commonly found in animal products) support the prevention and management of nutrition-related chronic disease conditions [11]. Principles of plant-based eating involve heavy emphasis on whole grains, fruits, vegetables, beans and legumes, nuts and seeds, herbs, spices, and plant oils, with limited consumption of meat and meat products [12]. Such sustained eating patterns have shown success in the reduction of obesity, improvement in obesity-related diseases, and provide a healthy variation to the typical Western or American-style diet [13]. Legumes and beans are recognized as both a vegetable and as a meat alternative because of a comparable nutrient profile that is high in protein, iron, and zinc [11,14,15]. *Phaseolus vulgaris*, common dry beans, are widely available, economical, versatile, and highly nutritious. Beans are a low glycemic source of complex carbohydrates, vitamins, minerals, protein, fiber, resistant starch [11,14,16,17,18], and phytochemical compounds with a multitude of bioactive properties [15]. The dietary inclusion of pulses, beans, and legumes is beneficial to improving disease conditions such as CVD and T2D [19], gut microbial diversity, colon health [20,21], and chronic low-grade inflammation [22]. Plant-based dietary patterns that include beans and legumes improve management of body weight, possibly reducing effects and consequences of obesity related to both non-communicable and communicable disease risk.

High blood pressure, high blood cholesterol, diabetes, and obesity are among key risk factors for heart disease [1,23]. Numerous studies have explored plant-based dietary patterns, as well as specific foods and components, to identify the role of key nutrients in the prevention, protection, and reversal of cardiometabolic diseases. Plant-based eating routines, such as the vegetarian or “flexitarian” diet, focus on the concept of zero to little meat and/or animal products, with the main intake of food coming from plant sources. Other widely recognized diets include the Mediterranean Diet, Healthy Mediterranean-Style Pattern, and Dietary Approaches to Stop Hypertension (DASH) Diet, which also heavily emphasize whole grains, fruits, vegetables, beans and legumes, nuts and seeds, herbs, spices, and plant oils, with inclusion of lean animal protein sources in reduced quantity [11,13]. There are additional variations that include the Mediterranean DASH Diet and the Vegetarian DASH Diet, among others [12]. A 2019 national survey of over 2000 adults found 4% to be vegetarian (including vegan) and 46% choosing to eat vegetarian or vegan sometimes, or always, when eating meals away from home [24]. Although there are accepted standards and characteristics that categorize these diet types, much variability in food selection and amount consumed remains within different cohorts, cultures, and individuals. Diets that consistently contain more plant foods and lesser amounts of animal foods are associated with less risk and incidence of type-2 diabetes, coronary artery disease, and mortality risk [13].

This review article provides an overview of the benefits of plant-based eating, with a concise focus on the nutritional properties unique to dry beans and their connection to improved health parameters of obesity including cardiovascular, metabolic, gastrointestinal gut health, and low-grade inflammation.

## 2. Plant-Based Dietary Patterns

### 2.1. Beans and Legumes in the Diet

The legume family, Fabaceae (Leguminosae), is made up of three subgroups that include oilseed legumes, fresh legumes, and pulses. Legume refers to plants that have their seeds enclosed in a pod. Soybeans and peanuts are considered oilseed legumes, while fresh legumes include beans and peas. Pulses are the dried seeds of legumes and include chickpeas, dried peas, lentils, and dry common beans (*Phaseolus vulgaris*) [25,26]. The terms “seeds”, “dried beans”, and “pulses” refer to the edible portion of the legume and are therefore often used interchangeably. Dry beans are a staple in the diets of many cultures in both industrialized and developing countries, comprising a significant portion of daily caloric intake. Common beans are widely available, inexpensive, and traditionally associated with frugality and food insecurity [27]. As the sixth-leading producer of dry beans, U.S. farmers annually plant and harvest 1.8 to 2 million acres of beans, including pinto, navy, great northern, red kidney, and black beans [2]. Consumption of legumes in the U.S. (beans, peas, lentils, and chickpeas) increased from 8 to 11.7 pounds per capita between 2014 and 2017, with peas, lentils, and pinto beans accounting for the highest demand. Black bean consumption from 1970 to 1990 was near zero but began to increase in popularity over the next several decades to 2.8 pounds per capita, placing them in third to last place from the bottom of the bean hierarchy [28].

The 2015–2020 Dietary Guidelines for Americans (DGA) includes 12 categories of food intake guidelines ranging from 1000 to 3200 calories, with further recommendations based on age and gender. Based on a typical 2000 calorie/day diet, 2.5 cup/equivalents (c-eq) of total vegetables should be consumed daily, including weekly recommendations within subcategories of dark-green, red/orange, starchy, legumes, and other vegetables. Under the Healthy U.S. and Mediterranean-Style Eating Patterns, the average adult male intake of legumes (beans and peas) is recommended in the range of 2 to 2.5 c-eq/wk, with female adults at 1.5 c-eq/wk. To meet protein requirements, an additional 6 oz-eq/wk of legumes and 8 oz-eq/wk soy products should be consumed for those following a 2000 calorie/day Healthy Vegetarian Eating Plan [11]. It is important to note that although peanuts and soybeans are legumes as well, they are categorized as oilseeds due to their high oil content and therefore are not considered to be pulses, grain legumes, or pulse grains [29,30,31]. Pulses, including all bean varieties, are rich in carbohydrate, resistant starch [16,17,18], fiber, potassium, copper, phosphorus, manganese, iron, magnesium, and B-vitamins, contain almost no sodium or fat, and are an excellent source of protein, with 21–25% content by weight [14]. Fat content of most *Phaseolus vulgaris* common dry bean varieties is very low, averaging 0.5 g total fat per half cup serving [14]. Additionally, the fatty acid content of dry beans such as navy, kidney, and black beans is favorable, with a low omega-6 (*n*-6; linoleic acid) to omega-3 (*n*-3; α-linolenic) ratio [32]. Although linoleic acid has been linked to obesity, in general, dry beans are overall low in linoleic acid content, with the exception of soybean and peanuts, which are not being discussed in this review [33,34]. Dietary consumption of pulses is associated with reduced risk of obesity [35], weight loss [36], and improved satiety [31,36]. Additionally, legumes are a source of rich bioactive phytochemical compounds that act in a number of metabolic and physiological processes, as well as exert a protective role that promotes chronic disease prevention [31,37]. The health promoting effects of beans and legumes is shown in Figure 1.

### 2.2. Nutritional Components of Beans

#### 2.2.1. Carbohydrate, Fiber, Resistant Starch

Beans are an inexpensive yet nutrient-dense food as they are a rich source of bioactive compounds including polysaccharides, oligosaccharides, protein, polyphenols/phytochemicals, and several vitamins and minerals. Pulses are mainly comprised of carbohydrate, making up over half of the total weight, between 55 and 65%. As the main form of carbohydrate, starch content in pulses includes a high amount of resistant starch (RS), as reflected in 30 to 40% amylose content and insusceptibility to digestion. In contrast, amylose makes up 15 to 30% of grain products such as cereal, allowing for elevated digestibility and blood glucose response rate [31]. Resistant starch moves through the digestive tract to the large intestine, lending itself to microbial fermentation processes and the production of beneficial short-chain fatty acids (SCFAs) [16,17,18]. The dietary fiber component of beans and pulses is very high, with some variation between types. For example, one serving equal to ½ cup cooked kidney beans has approximately 5.7 g of dietary fiber, while pinto beans have 7.7 g, and navy beans have 9.5 g [14]. Dietary fiber recommendations set forth by the DGA call for 14 g per 1000 kcal intake, or 28 g of fiber for a 2000 calorie diet [11]. Soluble and insoluble fiber, including RS, improve biologic mechanisms of cardiometabolic risk factors including glycemic and blood pressure control [38,39] and support a healthy gut bacterial environment [16,18]. With low average dietary fiber intake, most American adults consume an estimated 16 g per day, with more than 90% falling short of adequate intake (AI) standards [40]. Because of their elevated fiber concentration, even a small increase in bean consumption could allow notable improvement to fiber and RS profiles. Table 1 shows the macronutrient content of popular common bean varieties [14].

#### 2.2.2. Protein and Amino Acids

A ½ cup serving of cooked beans provides up to 25 g of protein, or approximately 20% of the recommended adult requirement [15]. Although protein content varies in different bean types or cultivars and is influenced by the growing environment [41], it is comparable to that of meat, with a 25% average caloric content [15,17]. Excessive consumption of meat products is associated with elevated fat and cholesterol intake and obesity. However, plant-based eating patterns provide a healthy alternative to the typical American-style diet [31]. Nutritional quality of food proteins is typically decided by essential amino acid composition and level of digestibility [42]. Protein digestibility and absorption is affected by the presence of active enzyme inhibitors in pulses. However, heat enables proteins to be denatured and hydrolyzed so that soaking and cooking beans increases protein and starch digestibility, thereby improving bioavailability and nutritional quality [31,43,44]. For instance, trypsin inhibitors in beans may be reduced by up to 90% by boiling [17]. As a rich source of protein, pulses contain notably high amounts of the essential amino acids lysine and leucine, with significantly lesser amounts of methionine and tryptophan [14,41,44]. Because most plant proteins are not complete in all amino acids, it is important to diversify daily intake to include other foods and ensure adequate intake of bioavailable protein. Assuming that calorie intake is optimal and a variety of plant foods are consumed, protein quality and content of U.S. vegetarian and vegan diets is typically sufficient [45].

#### 2.2.3. Vitamins and Minerals

The micronutrient content of legumes and pulses consists of an array of vitamins and minerals necessary to human nutrition. The vitamin profile of pulses includes vitamin C and seven out of the eight B-vitamins--thiamin, riboflavin, niacin, pantothenic acid, pyridoxine, biotin, and folate—but not vitamin B-12, which is primarily found in meat and animal protein sources. Beans are an excellent source of folate, providing roughly 70% of the recommended dietary allowance (RDA) per 1 cup serving [24,31]. Beans are also a source of niacin (B3), an important coenzyme responsible for the facilitation of many metabolic processes [31]. Additionally, the mineral composition is quite notable, with amounts of calcium, magnesium, phosphorus, copper, manganese, selenium, iron, zinc, and potassium [24], further supporting the high-quality nutritional profile of beans. Micronutrient composition is detailed in Table 2 [14].

#### 2.2.4. Phytochemical Components

The phytochemical nature of legumes, pulses, and beans is highly complex, comprised of a multitude of components that can greatly affect nutritive capacity and health. The bioavailability of several minerals is considerably lower for beans than from animal sources due to natural components, such as phytic acid, which inhibits and otherwise significantly interferes with the absorption of iron, zinc, and magnesium [46]. Although phytic acid concentration in pulses is lower than 2%, it is the main source of phosphorus [16]. Masum Akond et al. found an inverse relationship between phytic acid and iron, calcium, and magnesium concentrations but not zinc in a study of 29 bean varieties [46]. Although heat does not impair phytate function, effects may be decreased through methods of liquid soaking or fermentation, to result in improved zinc bioavailability and utilization [17].

Other interfering constituents, sometimes referred to as anti-nutrients, include oxalates and lectins. Oxalate disrupts the availability and absorption of pulse-derived calcium, resulting in only a fraction being utilized by the body during digestion [17,31,46]. If not adequately inactivated by cooking before consumption, high amounts of *Phaseolus vulgaris* lectins bind to the brush border of the small intestine, causing nausea, vomiting, and diarrhea, with potentially toxic effects including hyperblastosis and cellular necrosis [47]. On the other hand, bean lectins have been studied pharmacologically and demonstrated positive implications for immune response activation, cancer treatment--including liver, lung, breast, lymphoma, melanoma, and myeloma—as well as antiviral treatment for leukemia viruses, human immunodeficiency virus (HIV), and coronaviruses. Administered in extract form, phytohemagglutinin (PHA) from red kidney beans may stimulate important metabolic regulatory hormones for digestion and appetite satiation, with implications for obesity and T2D treatment through positive body weight, lipid, and glycemic management [47].

Pulses contain phytochemical components known as phenolic compounds, including flavonoids and phenolic acids, which exhibit a range of bioactive functions important to human health [15,37]. Phenolic compounds are responsible for the flavor and color of beans, varying in type and amount between different cultivars--they make up approximately 11% of a common dried bean and are concentrated mainly in the seed coating [15,37]. Many of these compounds, including flavonoids, provide seeds with natural protection against such pests as microbial invaders and insects [48]. The health benefits of polyphenols from common beans were recently summarized in a 2017 comprehensive review by Ganesan and Xu [15]. Studies that they examined demonstrated various bioactive effects of total phenolics, individual phenolic acids, flavonoids, anthocyanins, and tannins. Polyphenols are found to harbor cardioprotective, antidiabetic, and cancer-protective properties that exert antioxidant, anti-inflammatory, anti-hypertensive, hypolipidemic, hypoglycemic, anti-obesity, and antiproliferative activities [15]. Generally, darker beans, such as black beans, are a richer source of phytochemicals, with a higher antioxidant capacity, a higher phenolic content, and a more diverse profile of polyphenols, when compared to lighter beans [49]. Twelve varieties of *Phaseolus vulgaris* analyzed found the content of anthocyanins to be highest in black beans; black, red, as well as speckled beans demonstrated a positive antioxidant and antiproliferative correlation activity of high flavonoid and total polyphenol concentration [48]. Anthocyanins, mainly delphinidin, petunidin, and malvidin, are some of the most abundant phytochemicals found in black beans and have been shown to improve glycemic control and reduce the risk for cardiovascular disease [50]. Collectively, although beans possess anti-nutrients that are capable of impeding nutrient bioavailability, they also provide immune system support and cardiovascular, cancer, and metabolic protection through the bioactive capacity of lectins and phenolic compounds.

### 2.3. Role of Beans in Cardiometabolic Health

#### 2.3.1. Glycemic Control and Cardiovascular Disease

The Center for Disease Control and Prevention (CDC) estimates that 33.9% of the adult U.S. population (84.1 million individuals) are pre-diabetic, and of those individuals, 25% will likely develop type-2 diabetes (T2D) within three to five years, resulting in an estimated one in every three developing T2D by 2050 [51]. For individuals with T2D in the U.S., CVD is the leading cause of morbidity and mortality, and one in three men and two in five women with T2D will suffer cardiovascular-related complications or death [52]. The coexistence of pre-diabetes or T2D and CVD risk factors is not uncommon, and comorbid conditions including hypertension, dyslipidemia, obesity, and endothelial dysfunction (ED) may contribute to the development and progression of CVD [53,54]. Glycemic burden due to poor blood glucose control or impaired insulin response contributes to endothelial oxidative stress and serious health conditions including prediabetes, T2D, and CVD [38]. Recognized as a major plant-based protein, beans and legumes are an excellent source of complex carbohydrates and soluble fiber and tend to positively affect glucose control [38]. Epidemiological studies indicate an inverse association between the prevalence of chronic diseases such as T2D and the consumption of beans [55]. Legumes and beans favor low glycemic response when consumed alone, as part of low glycemic index (GI) diet, or diets higher in fiber and have been shown to improve both postprandial glycemic response and glycated hemoglobin (HbA1c) outcomes in T2D [38]. Phytochemicals in beans, such as anthocyanidins, lower postprandial blood glucose by inhibiting α-amylase, maltase, and sucrase activity [56]. Routine consumption of low-GI foods such as beans in an overall dietary pattern that decreases nutrition-related chronic disease risk [38]. It has been shown that combining beans (pinto, kidney, or black beans) with rice attenuates postprandial glycemic response in adults with T2D, when compared to rice alone [57]. The incorporation of traditional foods, such as common beans, which are high in fiber and phytochemicals, may help to improve conditions of hyperglycemia and dyslipidemia [58]. Beans have a low glycemic index, making them an ideal food for management of blood glucose and insulin resistance [59,60] as well as a positive influence on controlling blood pressure [61]. A 2012 study by Jenkins et al. demonstrated coronary heart disease (CHD) risk reduction and improved glycemic control in individuals with diagnosed T2D consuming a low-GI legume diet [19]. Including 190 g (1 cup) per day of legumes (cooked beans, chickpeas, or lentils) significantly improved HbA1c (−0.5% absolute), total cholesterol (−8 mg/dL), and triglycerides (−22 mg/dL), as well as reductions in blood pressure (−4.5 mm Hg systolic; −3.1 mm/Hg diastolic), heart rate (−3.1 bpm), body weight (−2.7 kg), and waist circumferences (−1.4 cm) with low-GI legume diet consumption compared to high wheat fiber diet [19]. Regular consumption of legumes/beans is an important behavioral dietary strategy to reduce risk and improve comorbidities of CVD and T2D.

#### 2.3.2. Vascular Health and Endothelial Dysfunction

Often a consequence of poor lifestyle behavior, chronic nutrition-related disease conditions, T2D, high blood pressure, and dyslipidemia, amplify the risk for the development of CVD [62]. Lifestyle interventions are the most important approaches for prevention of CVD. CVD is characterized as a group of heart and blood vessel disorders that includes coronary artery disease, cerebrovascular disease, and peripheral artery disease, among others [1]. Risk for CVD can be improved by healthy weight maintenance, increased physical activity, and consumption of an overall healthy diet [1,23]. Epidemiological outcomes from a large-scale U.S. study found regular consumption of legumes to be significantly associated with an 11% risk reduction in CVD and 22% in CHD when eaten at least four times per week, compared to less than once weekly [31,63]. Although genetics plays a part, lifestyle factors, health behaviors, and diet quality have major influences over the development and progression of both CVD and metabolic dysfunction [62].

Diet has a great influence on the risk of CVD, including dyslipidemia, lipotoxicity [64], and the development of atherosclerosis--a disease largely characterized by the accumulation of fatty lesions, resulting in plaque formation within the intima of arterial walls, which disrupts the normal functionality of the endothelial membrane [65]. As an important independent indicator of CVD risk, dyslipidemia is characterized by high plasma triglycerides (TG), a shift towards small dense low-density lipoprotein cholesterol (LDL) particles, and low high-density lipoprotein cholesterol (HDL) [53]. Several studies demonstrate the benefit of legumes in the reduction of elevated serum cholesterol levels. Meta-analysis of 10 studies of high intake of non-soy legume consumption found serum cholesterol improvements in total cholesterol (−11.76 mg/dL), LDL cholesterol (−7.98 mg/dL), triglycerides (−18.94 mg/dL), and HDL cholesterol (0.85 mg/dL) [66]. A cohort of adults over the age of 50 showed that a 2-month diet consisting of two equivalent servings daily of pulses (150 g wet weight, cooked) significantly decreased LDL cholesterol by 7.9% and total cholesterol by 8.3%, reducing CHD risk by as much as 25% [67]. Borderline high cholesterol participants experienced a 6% reduction in total cholesterol and a clinically significant reduction in LDL-C, potentially eliminating the need to receive pharmacological statin treatment [67]. Intake of legumes above the suggested guideline amounts indicates positive potential for vascular health and the treatment of cardiovascular risk conditions [22,67]. Hermsdorf et al. demonstrated significant reductions in total cholesterol concentrations and systolic blood pressure in a 2-month hypocaloric legume-based diet (4 servings per week) study of obese adults [22]. Additionally, study participants that consumed four servings per week of cooked legumes experienced a higher weight loss than the legume-restricted group and significantly higher percent reductions in pro-inflammatory marker C-reactive protein (CRP) concentrations [22].

Lipotoxicity is caused by chronic excessive levels of circulating lipids that result in oxidative stress and inflammation and extends to functional damage of vascular tissue and pathological organ damage [64]. The resulting endothelial dysfunction causes thickening of arteries, leading to CVD, the dominant cause of death in the United States [65]. Obesity, in combination with insulin resistance, diabetes, dyslipidemia, and lipotoxicity, is implicated in cardiomyopathy development and partly responsible for at least a two-fold increased risk for heart failure of obese individuals compared to those of healthy weight [64]. The endothelium comprises the inner cellular lining of blood vessels throughout the body, including arteries, veins, and capillaries, and plays an important role in many physiological activities in the body, including platelet aggregation, vascular tone, cell fluidity, and inflammation [68]. Endothelial dysfunction involves complex dynamics that trigger and sustain stiffening of the large arteries due to structural changes such as degeneration of elastin, increased collagen, expansion of the arterial wall, endothelial damage, and decrease in nitric oxide (NO) release, resulting in impaired vasodilation. Low-grade chronic inflammation, or metabolic inflammation, as a result of lipotoxicity and obesity-induced insulin resistance produces pro-inflammatory cytokines that, if left unresolved, tend to develop into extenuating comorbidities [64,69]. The relationship between obesity, chronic inflammation, and nutrition-related chronic diseases is shown in Figure 2. Markers of low-grade inflammation have been observed following the consumption of a meal high in fat and rapidly digestible carbohydrates in obese [70] and normal-weight [71] individuals. In rodents [72,73] and humans [74], meals of this composition have significant pro-inflammatory effects, further exacerbating vascular endothelial dysfunction through alterations in NO, excessive reactive oxygen species (ROS) production, and nuclear factor-κB (NF-κB) activation. Individuals with T2D may have endothelial dysfunction, arterial stiffness, and increased pro-inflammatory cytokines [55,61], which increase the risk for congestive heart failure, myocardial infarction, peripheral vascular disease, and stroke [49]. Some of the underlying mechanisms may include increased levels of circulating free fatty acids, visceral adiposity, activation of the renin-angiotensin aldosterone system, and increased circulating glucose [61]. These mechanisms contribute to oxidative stress, inflammation, and insulin resistance, which can ultimately result in a cardiovascular event [61]. Bioactive components, including phenolic compounds found in legumes, pulses, and beans, play a role in modulating vascular integrity and inflammatory markers. Atherogenic mice supplemented with a fresh-ground bean protein hydrolysate, equivalent to approximately one daily serving of cooked beans, showed significant reductions in plasma triglycerides and total cholesterol after nine weeks [75]. Furthermore, marked improvements in inflammation and endothelial dysfunction demonstrated 62% increased endothelial nitric oxide synthase (e-NOS) and 57% nitric oxide serum concentration, in addition to gene expression changes in TNFα (94% reduction) and angiotensin II (79% reduction), as compared to atherogenic diet alone. These vasodilatory outcomes are postulated to result from endothelial protective mechanisms of phytochemical and bioactive peptides in preventing oxidation of LDL and limiting activity of angiotensin II expression, enhancing the availability of NO via inhibition of angiotensin converting enzyme (ACE) [75]. Altogether, the inclusion of beans in the diet improves body weight and supports healthy vasculature by mechanisms that counteract dyslipidemia, high blood pressure, oxidative stress, and inflammation.

#### 2.3.3. Microbiome and Cardiometabolic Health

Interestingly, some research indicates that the adverse microbial composition of the gut is associated with obesity, insulin resistance, and T2D and cardiovascular disease risk factors [21,76,77]. More than 70% of human microorganisms reside within the gastrointestinal tract, providing a beneficially mutual, or commensal, relationship with the host. Although the composition of a “heathy gut” may vary greatly between individuals as it is influenced by environmental and hereditary factors, the phyla of *Bacteroidetes* and *Firmicutes* comprise over 90% of the total bacterial population [76,78]. The ratio and diversity of bacterial populations has been implicated to play a role in disease development, where a decreased *Bacteroidetes* to *Firmicutes* ratio and decreased overall microbial diversity may be linked to hypertension, obesity, and insulin resistance [76].

Diets that are high in fat are linked to insulin resistance and chronic low-grade inflammation. This is due in part to the negative influence of high-fat diet on microbiome composition, including gut permeability and pro-inflammatory cytokine production [76]. On the other hand, prebiotic compounds, including dietary fiber found in beans, benefit human health by providing protection against the development of certain diseases potentially via modulation in gut microbial composition [79,80,81,82,83,84,85,86,87]. Short-chain fatty acids (SCFAs) are formed from bacterial metabolism of dietary fiber and non-digestible resistant starches in the colon. The intestinal bacteria *Firmicutes* are responsible for producing butyrate, and *Bacteroidetes* produces mostly acetate and propionate [76]. SCFAs serve as signaling molecules and activators of GPR_41_ and GPR_43_, which may play an important role in improving glucose tolerance by regulation of energy balance mechanisms [76,82]. Increased production of butyrate exhibits an anti-obesity benefit and has been shown to contribute to reduction in body weight, body fat, and improved insulin sensitivity [21]. Inclusion of beans in the diet is shown to increase SCFAs and improve indicators of obesity-related disease conditions. In a 2-month study, rats fed black beans showed significant positive differences compared to control groups by exhibiting higher beneficial microbial diversity, increased abundance of butyrate-producing bacteria and elevated concentrations of butyrate, lowered rates of lipopolysaccharide (LPS) and intestinal NF-κB, significantly lower post-prandial serum glucose and insulin signaling, and a 28% reduction in body fat [20]. Interventions to modulate the gut microbial composition, including increased consumption of legumes, pulses, and beans, may be an effective treatment strategy to improve insulin resistance, inflammation, and comorbid conditions of obesity and T2D.

### 2.4. Role of Beans in Gut Health

#### Short Chain Fatty Acids, Inflammatory Response, and Cancer

Dysbiosis of the gut microbiome has been linked to several diseases including diabetes, cardiovascular disease [76], inflammatory bowel disease (IBD), and colorectal cancer (CRC) [83]. The relationship between cancer and modifiable risk factors, including diet, has been well established. Ranked as the third leading cause of cancer in the U.S. for adults, at least 45% of all cancers of the colon and rectum--collectively referred to as colorectal cancer (CRC) --may be attributed to behavioral factors [84], including excess food intake and obesity [21,85]. Diets devoid of fiber not only have a negative impact on gut microbiome diversity [86], but they encourage more susceptible degradation of the mucous membrane in the colon and potentially increase the risk for infection and disease [87]. Combination resistant starch and soluble fiber in beans allow for fermentation to occur in the distal colon, driving increased production of short-chain fatty acids (SCFAs) by microbiota in the human body, particularly butyrate [21]. Butyrate, an SCFA produced from non-digestible components, serves as an important source of energy for supporting mechanisms of gut health [18,83,85,88,89]. Butyrate is produced as a result of bacterial conversion of acetyl-CoA to butyryl-CoA, which is then absorbed by colonocytes, where it is directly involved with disease repression mechanisms of inflammation and cancer, including histone deacetylase (HDAC) inhibitory action in the modulation of cancer-promoting genes [18,83,85,88,89]. Hangen and Bennink found that rats exposed to colon carcinogen azoxymethane and fed a diet of black beans or navy beans during a 36-week experimental period had over 12% lower body weight, significantly reduced incidence of colon adenocarcinomas (44–75%) and total tumors (54–59%), and colonic butyrate concentrations nine times higher than control groups [21].

SCFAs butyrate, acetate, and propionate from resistant starch benefit colon health through multiple mechanisms that favor an environment to support healthy colonocyte activity, growth, and development [16,17,18]. Mice fed black beans or navy beans exhibited improved mucus epithelial barrier integrity and decreased permeability in conjunction with increased microbial carbohydrate fermentation and reduced protein fermentation, as evidenced by elevated SCFAs and decreased branched-chain fatty acids (BCFAs) [49]. Additionally, SCFAs have great potential for reducing adverse events which develop from chronic dysregulated inflammatory response mechanisms, which are the hallmark of gastrointestinal conditions such as IBD, colitis, and colon cancer [18]. Host microbiota are instrumental participants in inflammatory mechanisms that either encourage or diminish IBD and cancerous tumor development [83]. Lack of diversity and ratio inequality of gut microbiota have been reported in CRC patients, with significantly abundant pathogenic bacteria such as *Streptococcus* and *Escherichia-Shigella* and lesser amounts of butyrate-producing genera including *Roseburia* [83]. Those that suffer with IBD exhibit negative microbiome diversity and may have an average of 25% fewer fecal microbial gene types than typical healthy-gut individuals [83]. In a mouse model of colitis, Zhang et al. tested the effects of bean flour (whole cooked, freeze-dried) in modulating initial colon disease severity, demonstrating that black bean and navy bean supplementation improves markers of inflammation in colitis by mechanisms that involve bioactive factors, phenolic compounds, and gut fermentable components [18]. Although bean diets enhanced phenolic profile and SCFA concentration through reduced gene expression of several key pro-inflammatory cytokines, upregulation of anti-inflammatory mediator IL-10, and improved local inflammation of the colon, interestingly, they also increased and prolonged colon tissue damage and apoptosis [18]. Bean matrix complexity and downstream mechanisms that involve the gut bacterial environment support a somewhat codependent relationship that can improve underlying factors of chronic disease states.

### 2.5. Plant-Based Dietary Strategies May Improve Obesity, Immune System, and Modulate COVID-19 Risk

The potential implications of COVID-19 are severe for individuals who are obese, metabolically compromised, and/or have heart disease [7,8]. The recent global transmission of severe acute respiratory syndrome coronavirus-2 (SARS-CoV-2) has led to a massive pandemic, with which the U.S. accounts for over 20% of diagnosed cases [8,90].

Anyone that has been exposed to the virus is at risk for contracting it and spreading it to others; however, there is evidence that older people and those with certain underlying health conditions are at an increased risk of devastating disease severity [8]. Medical conditions include cancer, chronic kidney disease, COPD, coronary artery disease, T2D, and obesity. Those with metabolic and cardiovascular disease conditions have an increased likelihood of becoming hospitalized if COVID-19 is contracted [8]. The CDC has indicated a three-fold increased risk of hospitalization for diabetic or obese (BMI ≥ 30) people and a 4.5-fold increase for severely obese conditions with BMI ≥ 40 [8]. Even more alarming, individuals with multiple health conditions may be nearly five times more likely to require hospitalization and advanced medical care [8]. For some, case severity may escalate enough to necessitate intense medical treatment and care, ultimately culminating in acute respiratory distress syndrome (ARDS), sepsis, multiple organ failure, and death [9,91].

Obesity is characterized as an excess accumulation of adipose tissue and is a modifiable risk factor associated with CVD, hypertension, and T2D [92]. Body mass index (BMI) and Gross Domestic Product (GDP) have been identified as strong predictors of COVID-19 death in a retrospective correlation study released in May 2020 [7]. Rates of obesity in the U.S. continue to climb, with over 42% of all adults having a BMI ≥ 30, including over 9% categorized as severely obese [3]. It is surmised that because countries with higher incomes have innumerable food resources, dietary intakes comprise more animal products, such as meat and dairy, and less plant-derived sources, directly impacting the rate of obesity [7]. On the other hand, environmental and socioeconomic factors also play a large role in health inequity with the U.S. and behaviors that support obese conditions within subsets of populations and communities [93].

Obese patients are at an increased risk of COVID-19 complications, particularly if comorbidities such as T2D or CVD are involved [8,9,10]. Dyslipidemia is often seen in obesity and T2D and is a risk factor for CVD. A meta-analysis of nearly 7000 patients found a significant association between dyslipidemia and COVID-19 disease severity, indicating that high levels of LDL and low levels of HDL may play a role in dysregulated immune response and cardiovascular complications [94]. The body’s inflammatory response to coronavirus-2 (SARS-CoV-2) infection seems to play a key role in the onset, severity, and outcome of COVID-19 disease [9,91]. Referred to as “cytokine storm”, rapid and aggressive influx of pro-inflammatory cytokines by the immune system, in some people, overreacts in an uncontrolled manner. This hyper-response may result in pathological pulmonary tissue damage, impaired capacity of the lungs, acute respiratory distress, sepsis, and subsequent multiple-organ failure [9,91]. Additionally, COVID-19-induced blood vessel changes increase the incidence of hypercoagulation, thrombosis, and thromboembolisms [9]. Obesity is associated with chronic increased cytokine production and low-grade inflammation [7,10], putting these individuals at a greater risk for experiencing cytokine storms [91]. This may be due in part to immune system malfunction and an impaired pro-inflammatory response to viral infection, exacerbating complications of COVID-19 [91].

Often a reflection of excess calorie consumption, obese individuals typically have poor eating habits that include high intakes of refined starch, saturated fat, and added sugar [7,92]. Quarantine during COVID-19 pandemic has forced many to conduct their lives on a virtual platform from home, greatly impacting the ability to move about freely outside of the home environment. This change in lifestyle may negatively affect behaviors related to food intake and physical activity, potentially increasing the prevalence of overweight and obesity cases. A small cohort of obese residents in Italy were reported to have significant weight gain after just one month of quarantine [95]. Behaviors associated with the most significant body weight and BMI increases include anxiety/depression and lack of mindfulness in food selection. Other unhealthy dietary habits include consuming higher quantities of food, particularly snacks and sweets, and less fruits and vegetables than before lockdown [95]. In the U.S., excessive consumption of heavily processed foods and diminished adherence to U.S. Dietary Guidelines may be largely responsible for common nutrient deficiencies associated with overweight populations [92], further amplifying obese conditions. Nutrients of concern include vitamins A, D, E, and C; calcium, magnesium, phosphorus, zinc, iron, and fiber [92]. Micronutrient deficiencies in obese individuals may have deleterious effects, particularly in relation to chronic disease prevention, inflammation, effective immune response, disease duration, and severity.

Nutrition is a crucial factor in moderating the balance between healthy and disease states. Bioactive phytochemicals from whole plant foods provide enhanced benefits above macro- and micronutrient content. Phenolic compounds and resistant starch content of common beans are among characteristics that make them unique, supporting their capacity to prevent, mitigate, and protect against human disease. For instance, butyrate from increased and sustained consumption of black beans and navy beans is shown to improve mucus epithelial barrier integrity, decrease gut permeability [49], as well as modulate gene expression of key pro-inflammatory cytokines in the colon and circulating serum concentrations [18]. Interestingly, mechanisms involving SCFAs and their action on G-protein-coupled-receptors (in gut epithelium, adipose tissue, and immune cells) may also serve to improve overall respiratory health and mitigate potential tissue damage of viral disease acquisition [82,96]. Perhaps risk-reduction strategies that include healthy body weight and improved nutritional status should be considered for combatting exposure to SARS-CoV-2 in obese individuals. In particular, focus on a plant-based diet with increased intake of beans and legumes would be advantageous for many reasons previously stated, but also as a protective measure that serves to boost the body’s defenses in an acute immune response.

## 3. Discussion

Lifestyle factors, health behaviors, and diet have major influence over obesity and the incidence of CVD and metabolic dysfunction [62]. High amounts of dietary fiber, antioxidant vitamins and minerals, phytonutrients, and Omega-3 fatty acids--as well as potentially their combined effect--that make up the Mediterranean Diet, and other plant-based diets, are profoundly effective in the prevention and modulation of obesity-related chronic diseases including CVD and T2D [62,93]. For instance, phytochemicals found in beans and legumes are considerably beneficial in improving blood cholesterol levels, glycemic status, providing vascular protection, and reducing markers of chronic inflammation [62]. Furthermore, SCFAs produced from the fermentation of complex dietary fiber and resistant starches in beans are important in supporting healthy gut microbial population and diversity. Increased butyrate production positively influences the gut microbiome in the reduction of body weight, body fat, and improved insulin sensitivity [21]. Notwithstanding the degree and importance of optimal nutrient intake necessary to prevent and combat obesity-related diseases such as CVD, T2D, and cancer it is also advantageous for the functionality and support of the body’s “home security” defense system against microorganism invasion. Plant-based dietary approaches that support optimal nutrient intake, healthy body weight, and reduced inflammatory status may be an effective protective force against immune-related diseases.

## Figures and Tables

**Figure 1 nutrients-13-00519-f001:**
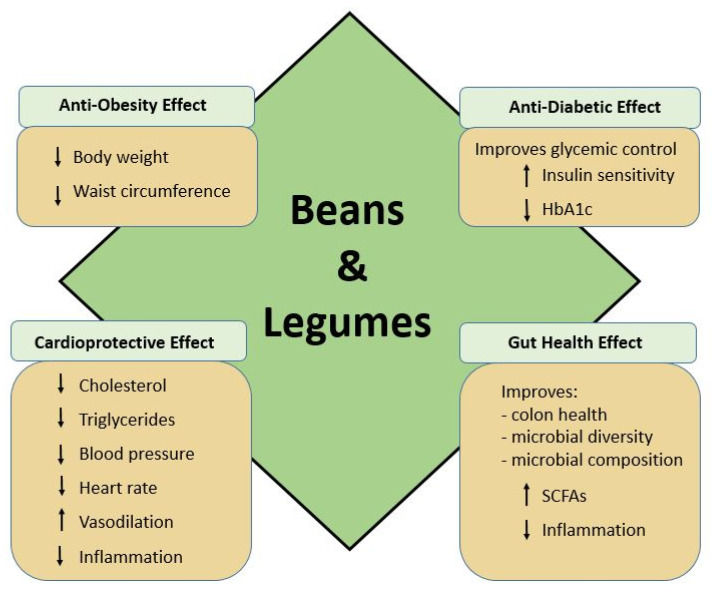
Health benefits of beans and legumes.

**Figure 2 nutrients-13-00519-f002:**
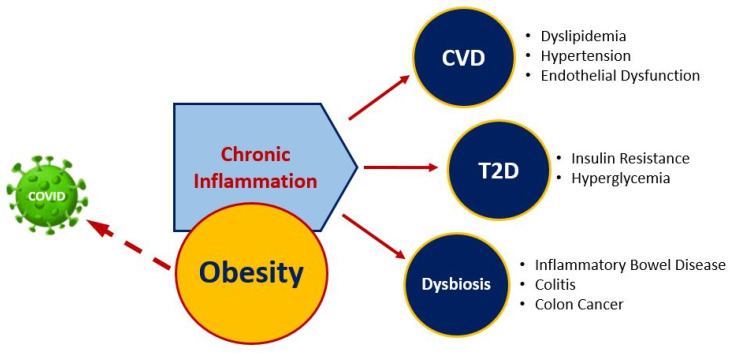
Obesity and nutrition-related chronic diseases.

**Table 1 nutrients-13-00519-t001:** Macronutrient content of common beans in 100 g of edible portion.

Nutrient	Units	Form	Pinto	Navy	Great Northern	Red Kidney	Black
Energy	kcal		143	140	118	127	132
Total Lipid	g	dry cooked	1.24 0.65	1.51 0.62	1.24 0.45	1.31 0.5	1.45 0.54
Protein	g	dry cooked	23.7 9.01	24.1 8.23	24.7 8.33	25.9 8.67	24.4 8.86
Total Carbohydrate	g	dry cooked	n/a 26.22	n/a 26.05	n/a 21.1	n/a 22.8	n/a 23.71
Dietary Fiber	g	dry cooked	4.1 9.0	4.3 10.5	4.3 7.0	4.3 7.4	4.2 8.7
Starch	g	dry cooked	38.5 15.15	38.5 15.15	37.9 n/a	36.7 n/a	36.6 n/a

**Table 2 nutrients-13-00519-t002:** Micronutrient composition of common beans in 100 g of edible portion.

Nutrient	Units	Form	Pinto	Navy	Great Northern	Red Kidney	Black
Calcium	mg	dry cooked	161 46	229 69	192 68	98 28	191 27
Iron	mg	dry cooked	5.4 2.09	5.29 2.36	180 53	523 144	5.34 2.10
Magnesium	mg	dry cooked	170 50	180 53	176 50	164 45	180 70
Phosphorus	mg	dry cooked	507 147	523 144	519 165	612 142	522 140
Potassium	mg	dry cooked	1510 436	1470 389	1520 391	1490 403	1540 355
Zinc	mg	dry cooked	3.43 0.98	3.31 1.03	3.45 0.88	3.29 1.07	3.37 1.12
Copper	mg	dry cooked	0.978 0.219	1.140 0.210	1.080 0.247	0.865 0.242	1.120 0.209
Manganese	mg	dry cooked	1.78 0.453	2.22 0.527	1.90 0.518	1.67 0.477	2.08 0.444
Folate	μg	n/a	172	140	102	130	149

## Data Availability

Not applicable.

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
