# Peer review of "Health Benefits of Plant-Based Nutrition: Focus on Beans in Cardiometabolic Diseases"

_nutrients, 2021, doi:10.3390/nu13020519_

Round 1
Reviewer 1 Report
Re: Manuscript #nutrients-1062659, Title: Health Benefits of Plant-Based Nutrition: A Focus on Beans in Obesity-Related Diseases
Summary
In this review manuscript, Mullins and Arjmandi addressed the beneficial effect of plant-based nutrition, especially beans and legume in cardiovascular diseases based on the improvement of its risk factors and coexisting symptoms including obesity, type 2 diabetes and low-grade inflammation. In addition to those diseases/symptoms, the authors further addressed the improvement of immune system which may contribute to prevent the current pandemic burden, COVID-19 infection.
This manuscript itself is well written. Unfortunately, however, this manuscript should be considered as an incomplete and insufficient as a “complete” manuscript. No materials (figures and tables) attached. Word tracker residue can be found. Those factors encumber to get right information and read through. This issue had also made me to suspect that there is another manuscript which the authors revised by their side to consider as a final version to submit.
Therefore, I had read through and commented below for the unelaborated points.
Major comments
First of all, where are the tables and figures? Is this manuscript final version to submit?
The initial sentence of abstract and introduction, I wondered if the readers may get confused what was the main target for the authors. Although the title state “Obesity-Related Diseases”, both sentences have initiated with cardiovascular disease. It would be, I feel, better to initiate with obesity statement and then, move into its risk to develop cardiovascular diseases or else. It is understandable the authors want to emphasize the deadliest diseases to give impact. Therefore, I suggest to authors either the modification of the structure of sentences or the modification of the title. For example, replacing the “Obesity-Related Diseases” with “Cardiometabolic Diseases (or health)”.
From Line 68 to Line 84 seems general introduction of section 2. I suggest the authors to move before subsection title “2.1. Beans and Legumes in the Diet”.
Section 2.2.1: it would be helpful to show the nutrition facts as a table for some of major beans. Since there is no Table1, I’m not sure if that information is included in table 1.
Line 266: The reference #55 is not proper citation. Please replace with more relevant reference(s).
Minor comments
Reference #5: The authors cite this paper as a paper to explain about the obese condition in inflammation or some other symptoms in obese patients and its correlation to COVID-19 (Line 41-42 or elsewhere in the manuscript). However, the main theme of this paper is the correlation of GDP and death ratio around the world. They have cited some papers related to the obesity influence in the inflammation and COVID-19 pandemic. I assume there are the other papers much more suitable for each paragraph in this manuscript.
I would recommend replacing “cardiovascular and metabolic disease” with “Cardiometabolic diseases” in line 71.
Line 85, “cardio-protective” should be “cardioprotective”.
Line 325-326: The sentence “Interestingly, research indicates that adverse microbial composition of the gut is associated with obesity, insulin resistance and T2D, and cardiovascular disease risk factors [67,68,20].” should be “Interestingly, some researches indicate that the adverse microbial composition of the gut is associated with obesity, insulin resistance and T2D, and cardiovascular disease risk factors [67,68,20].”. Here I just pointed out one example. Please check through the manuscript about the singular/plural forms of verbs, prepositions and the necessity of “the”.
Author Response
Overall Response: We are grateful to the reviewers for the comments and suggestions that helped us to enhance the quality of this revised manuscript considerably. We have made all of the suggested changes to the manuscript and have individually responded to each reviewer.
Reviewer 1:
1) First of all, where are the tables and figures? Is this manuscript final version to submit?
Response: Our apologies, tables and models are now included in the body of the manuscript.
2) I suggest to authors either the modification of the structure of sentences or the modification of the title. For example, replacing the “Obesity-Related Diseases” with “Cardiometabolic Diseases (or health)”.
Response: Thank you for this suggestion. We agree and have modified the manuscript title to “Health Benefits of Plant-Based Nutrition: Focus on Beans in Cardiometabolic Diseases”
3) From Line 68 to Line 84 seems general introduction of section 2. I suggest the authors to move before subsection title “2.1. Beans and Legumes in the Diet”.
Response: Thank you again, this is an excellent point. The paragraph in question has been relocated to the “Introduction” section, lines 61-77.
4) Section 2.2.1: it would be helpful to show the nutrition facts as a table for some of major beans. Since there is no Table1, I’m not sure if that information is included in table 1.
Response: Our sincerest apologies for not previously including a nutrition facts table. Please see Tables 1 and 2 for nutrient composition of common beans.
5) Line 266: The reference #55 is not proper citation. Please replace with more relevant reference(s).
Response: The reference in question has been changed to #63.
7) Reference #5: The authors cite this paper as a paper to explain about the obese condition in inflammation or some other symptoms in obese patients and its correlation to COVID-19 (Line 41-42 or elsewhere in the manuscript). However, the main theme of this paper is the correlation of GDP and death ratio around the world. They have cited some papers related to the obesity influence in the inflammation and COVID-19 pandemic. I assume there are the other papers much more suitable for each paragraph in this manuscript.
Response: More appropriate references have been cited and revision made for lines 39-42.
8) I would recommend replacing “cardiovascular and metabolic disease” with “Cardiometabolic diseases” in line 71.
Response: We appreciate this recommendation and have changed "cardiovascular and metabolic disease" to "cardiometabolic diseases" in line 64.
9) Line 85, “cardio-protective” should be “cardioprotective”.
Response: Changed "cardio-protective" to cardioprotective" in line 85.
10.) Line 325-326: The sentence “Interestingly, research indicates that adverse microbial composition of the gut is associated with obesity, insulin resistance and T2D, and cardiovascular disease risk factors [67,68,20].” should be “Interestingly, some researches indicate that the adverse microbial composition of the gut is associated with obesity, insulin resistance and T2D, and cardiovascular disease risk factors [67,68,20].”. Here I just pointed out one example. Please check through the manuscript about the singular/plural forms of verbs, prepositions and the necessity of “the”.
Response: Thank you for pointing this out. Lines 340-341 now reads “Interestingly, some researchers indicate that the adverse microbial composition of the gut is associated with obesity, insulin resistance and T2D, and cardiovascular disease risk factors [74,75,21]”.
Reviewer 2 Report
Comments:
The authors have not mentioned about the negative role of linoleic acid, a major constituent of legumes towards weight management. While the amount varies in different types of legumes, there should be a highlight about the amount of intake of these legumes. Intake of soybean oil has been linked to obesity. In this context, the authors must cite previous literatures as follows:
Dietary linoleic acidelevates endogenous 2-AG and anandamide and induces obesity. Alvheim et.al., Obesity,2012
Linoleic acid in diets of mice increases total endocannabinoid levels in bowel and liver: modification by dietary glucose, Ghosh et.al., Obesity Science and Practice, 2019.
A Comprehensive Characterisation of Volatile and Fatty Acid Profiles of Legume Seeds, Khrisanapant et.al., Foods, 2019.
The section: 2.2.3. Vitamins and Minerals does not contain information about the benefits of beans with regards to obesity. Please expand this section.
Please provide citation for this statement: If not adequately inactivated by cooking before consumption, high amounts of Phaseolus vulgaris lectins bind to the brush border of the small intestine causing nausea, vomiting, and diarrhea with potentially toxic effects including hyperblastosis and cellular necrosis.
Please provide reference to the following sentence : Often a consequence of poor lifestyle behavior, chronic nutrition-related disease conditions, T2D, high blood pressure, and dyslipidemia, amplify the risk for the development of CVD.
Please provide reference to the following sentence: Intake of legumes above the suggested guideline amounts indicate positive potential for vascular health and the treatment of cardiovascular risk conditions.
Please provide reference to the following sentence: Some of the underlying mechanisms may include 306 increased levels of circulating free fatty acids, visceral adiposity, activation of the renin-angiotensin 307 aldosterone system, and increased circulating glucose.
Please indicate the full form of HDAC
Please abridge the content in “Plant-Based Dietary Strategies May Improve Obesity, Immune System, and Modulate COVID-19 Risk”. Detail has been presented towards the pathology of COVID-19, please shorten it and only highlight why it is important to focus on plant based strategies for disease management of covid-19.
Please provide reference for the following statement: Those with metabolic and cardiovascular disease conditions have an increased likelihood of becoming hospitalized if COVID 19 is contracted. The CDC has indicated a 3 fold increased risk of hospitalization for diabetic, or obese (BMI > 30) people and a 4.5 fold increase for severely obese conditions with BMI > 40.
“Nutrients of concern include vitamins A, D, E, and C; calcium, magnesium, phosphorus, zinc, iron and fiber [83 471 ]. Micronutrient deficiencies in obese individuals may have deleterious effects, particularly in relation to chronic disease prevention, inflammation, effective immune response, disease duration and severity”.
In this context, does the intake of one kind of bean or a combination of beans is useful? What other food should be taken further to improve bioavailability of the nutrition of beans?
The authors must describe the recommended dose of different varieties of beans in different age groups and gender, based on published literatures for optimal functions.
Author Response
Overall Response: We are grateful to the reviewers for the comments and suggestions that helped us to enhance the quality of this revised manuscript considerably. We have made all of the suggested changes to the manuscript and have individually responded to each reviewer.
Reviewer 2:
1) The authors have not mentioned about the negative role of linoleic acid, a major constituent of legumes towards weight management. While the amount varies in different types of legumes, there should be a highlight about the amount of intake of these legumes. Intake of soybean oil has been linked to obesity. In this context, the authors must cite previous literatures as follows:
Dietary linoleic acid elevates endogenous 2-AG and anandamide and induces obesity. Alvheim et.al, Obesity,2012
Linoleic acid in diets of mice increases total endocannabinoid levels in bowel and liver: modification by dietary glucose, Ghosh et.al, Obesity Science and Practice, 2019.
A Comprehensive Characterization of Volatile and Fatty Acid Profiles of Legume Seeds, Khrisanapant et.al, Foods, 2019.
Response: We thank the reviewer for bringing this matter to our attention. Now we have added a statement with reference to linoleic acid and citing the 3 references suggested (lines 111-116). We also included further description of this review regarding our focus on dry beans with the addition of lines 106-108.
2.) The section: 2.2.3. Vitamins and Minerals does not contain information about the benefits of beans with regards to obesity. Please expand this section.
Response: To address your comment, we have added lines 116-117 to the subsection 2.1 “Beans and Legumes in the Diet”.
3) Please provide citation for this statement: If not adequately inactivated by cooking before consumption, high amounts of Phaseolus vulgaris lectins bind to the brush border of the small intestine causing nausea, vomiting, and diarrhea with potentially toxic effects including hyperblastosis and cellular necrosis.
Response: We have cited the statement in lines 191-194 with reference #45.
4) Please provide reference to the following sentence: Often a consequence of poor lifestyle behavior, chronic nutrition-related disease conditions, T2D, high blood pressure, and dyslipidemia, amplify the risk for the development of CVD.
Response: We have cited the statement in lines 267-268 with reference #60.
5) Please provide reference to the following sentence: Intake of legumes above the suggested guideline amounts indicate positive potential for vascular health and the treatment of cardiovascular risk conditions.
Response: We have cited the statement in lines 292-293 with reference #65 and #22.
6) Please provide reference to the following sentence: Some of the underlying mechanisms may include 306 increased levels of circulating free fatty acids, visceral adiposity, activation of the renin-angiotensin 307 aldosterone system, and increased circulating glucose.
Response: We have cited the statement in lines 320-322 with reference #59.
7) Please indicate the full form of HDAC
Response: Revision made to Lines 384-385 to read “histone deacetylase (HDAC)”.
8) Please abridge the content in “Plant-Based Dietary Strategies May Improve Obesity, Immune System, and Modulate COVID-19 Risk”. Detail has been presented towards the pathology of COVID-19, please shorten it and only highlight why it is important to focus on plant based strategies for disease management of covid-19.
Response: We thank the reviewer very much for this suggestion. This section has been condensed with removal of this paragraph: “As climbing infection rates persist, extensive efforts to control the spread of the disease continue to be implemented through social behavioral measures and personal hygiene. These preventative practices include social and physical distancing, personal face coverings, effective handwashing, and frequent touch surface sanitizing, with further recommendations to avoid crowded situations especially in enclosed spaces [8]. Transmission of this highly infectious viral disease mainly occurs by person to person contact and through small respiratory droplet exposure (increasing with duration of exposure) from symptomatic or asymptomatic individuals [8]. Typical presentation of symptoms occur between 2 to 14 days. COVID-19 is primarily an acute respiratory illness with symptoms ranging from mild to severe and may include fever, cough, shortness of breath, congestion, fatigue, muscle aches, headache, mental confusion, loss of taste or smell, nausea, vomiting, or diarrhea [8]. Individuals may exhibit signs of debilitating illness, while others experience very mild discomfort often mistaken for common ailments. Even still, an estimated 35% of infected people remain asymptomatic, never exhibit indications of having the virus at all, yet are able to unknowingly spread SARS-CoV-2 through viral shedding [81, 8]”.
9) Please provide reference for the following statement: Those with metabolic and cardiovascular disease conditions have an increased likelihood of becoming hospitalized if COVID 19 is contracted. The CDC has indicated a 3 fold increased risk of hospitalization for diabetic, or obese (BMI > 30) people and a 4.5 fold increase for severely obese conditions with BMI > 40.
Response: We have cited the statement in lines 421-424 with reference #8.
10.) “Nutrients of concern include vitamins A, D, E, and C; calcium, magnesium, phosphorus, zinc, iron and fiber [83 471]. Micronutrient deficiencies in obese individuals may have deleterious effects, particularly in relation to chronic disease prevention, inflammation, effective immune response, disease duration and severity”. In this context, does the intake of one kind of bean or a combination of beans is useful? What other food should be taken further to improve bioavailability of the nutrition of beans?
Response: Again we thank the reviewer for bringing this to our attention. There is a statement in the Phytochemical Components section (lines 186-188) which explains a few things that one can do to reduce the impact of phytate and bioavailability of bean compositions.
11) The authors must describe the recommended dose of different varieties of beans in different age groups and gender, based on published literatures for optimal functions.
Response: This very important point has been addressed (lines 98-106) with the addition of information regarding weekly bean intake recommendations of three different eating patterns within the Dietary Guidelines for Americans, based on gender and age.
Reviewer 3 Report
Well written concise review of the role of legumes in improving metabolic health and decreasing the incidence and prevalence of metabolic and cardiovascular disease conditions. The link to the risk factors that lead to increased morbidity and mortality of COVID-19 pandemic is novel and we'll justified.
Author Response
Response to Reviewer Comments for Manuscript ID - 1062659
Overall Response: We are grateful to the reviewers for the comments and suggestions that helped us to enhance the quality of this revised manuscript considerably. We have made all of the suggested changes to the manuscript and have individually responded to each reviewer.
Reviewer 3:
Well written concise review of the role of legumes in improving metabolic health and decreasing the incidence and prevalence of metabolic and cardiovascular disease conditions. The link to the risk factors that lead to increased morbidity and mortality of COVID-19 pandemic is novel and we'll justified.
Response: We sincerely thank the reviewer for taking the time to read and provide comments on this manuscript.
Round 2
Reviewer 1 Report
Re: Manuscript # nutrients-1062659-peer-review-v2, Title: Health Benefits of Plant-Based Nutrition: A Focus on Beans in Obesity-Related Diseases
Summary
In this review manuscript, Mullins and Arjmandi addressed the beneficial effect of plant-based nutrition, especially beans and legume in cardiovascular diseases based on the improvement of its risk factors and coexisting symptoms including obesity, type 2 diabetes and low-grade inflammation. In addition to those diseases/symptoms, the authors further addressed the improvement of immune system which may contribute to prevent the current pandemic burden, COVID-19 infection.
The revised version of the manuscript seems fine with some materials which had lost in the previous version. Nice work.
Only few points for materials since there were no materials (tables and figures) in the previous manuscript.
Figure 1: The fonts for symptoms of light side is a bit small. Could you make the font bigger? Also, arrowhead pointing the COVID is also a bit small. Could you make it larger to make the message clearer?
Table 1: I personally prefer to put (/100g) at the end of table title, “Macronutrient Content of Most Commonly Consumed Bean Varieties in the US” and remove the column of Amount. The current version is still fine, though.
Table 2: The size itself is smaller than Table 1. Please make it consistent with the size (font) with table 1. If you remove the column of “Amount”, I believe it is possible.
Figure 2: It would be better to add a bit more detailed information. For example, in the cardioprotective part, “Cholesterol” to “Lowering cholesterol” or “Cholesterol↓”, or else.
By the way, for this kind of title, the adjectives are not common. “Cardioprotective” should be “Cardioprotection” or “Cardioprotective effect” would be natural. “Anti-Diabetic” and “Gut Microbiota supportive” are also the same.
The comments for the answers for previous comments are below.
Comment 1) OK. Now it seems complete “Manuscript”, not draft of Manuscript.
Comment 2) OK.
Comment 3) OK. By the way, I meant to move between the section title “2. Plant-Based Dietary Patterns” and subtitle “2. 1. Beans and Legumes in the Diet”. Although the current version itself is still fine, it would be better to place that sentences in there, at least for me, to move into subsection. It’s just personal preference. Either way would be fine.
Comment 4) OK
Comment 5) OK
Comment 7) OK
Comment 8) OK
Comment 9) OK
Comment 10) OK
Author Response

(The authors gave the same response as above.)
